# Assessment of anesthesia capacity for children in Somaliland

**Mubarak Mohamed**[1‡], **Andie Grimm**[2‡], **Christina Williams**[3], **Cesia Cotache-Condor**[4,5,6], **Tessa L. Concepcion**[7], **Shukri Dahir**[1], **Edna Adan Ismail**[8], **Henry E. Rice**[4,5,6], **Emily R. Smith**[5,6,9]*

1 Department of Surgery, Edna Adan University Hospital, Hargeisa, Somaliland, 2 Institute for Cancer Outcomes and Survivorship, University Alabama at Birmingham, Birmingham, Alabama, United States of America, 3 Program in Pediatrics and Anesthesiology, Boston Children's Hospital, Boston, Massachusetts, United States of America, 4 Department of Surgery, Duke University School of Medicine, Durham, North Carolina, United States of America, 5 Duke Center for Global Surgery and Health Equity, Duke Global Health Institute, Duke University, Durham, North Carolina, United States of America, 6 Duke Global Health Institute, Duke University, Durham, North Carolina, United States of America, 7 Department of Global Health, University of Washington, Seattle, Washington, United States of America, 8 Founder and Director, Edna Adan University Hospital, Hargeisa, Somaliland, 9 Department of Emergency Medicine, Duke University, Durham, North Carolina, United States of America

‡ MM and AG are shared first-authors on this work.
* emily.smith1@duke.edu

**Data Availability Statement:** Data is available upon reasonable request. The data is available upon request to the PI and the ethics board per the Duke IRB approval for this protocol. The Duke IRB

## Abstract

The burden of pediatric surgical conditions in Somaliland is high and the pediatric anesthesia capacity across the country remains poorly understood. The international standards developed by the World Health Organization and World Federation of Societies of Anaesthesiologists (WHO-WFSA) serve as a guideline to assess the provision of anesthetic care. This study aims to describe anesthesia capacity for children in Somaliland and assess progress towards reaching the WHO-WFSA international standards. In this cross-sectional study, anesthesia infrastructure and workforce data, as well as pediatric clinical and demographic data were collected from fifteen private, charity, and government hospitals in the six regions of Somaliland. We described anesthesia capacity in Somaliland and compared baseline data to the WHO-WFSA international standards. Overall, Somaliland did not reach most of the target goals for anesthesia capacity as defined by the WHO-WFSA. Most markers for anesthesia capacity were far behind the established targets, with deficits of 99% for anesthesiologists density, 83% for operating room density, and 83% for ventilator density. Hospitals in urban Maroodi-Jeex, and private hospitals had more supplies, infrastructure, and surgical personnel than hospitals in rural areas. There are large gaps in current anesthetic resources for children according to WHO-WFSA global standards, as well as wide disparities between regions and types of hospitals in Somaliland. Increased investment in anesthesia infrastructure and workforce is required to meet the needs of pediatric surgical patients across the country.

prohibits letting the data be made freely available in an open source repository or as supplementary material. However, persons can request access for the data as stated previously. Please contact Debbie Hickey at the Duke IRB for inquiries: deborah.hickey@duke.edu.

**Funding:** The author(s) received no specific funding for this work.

**Competing interests:** The authors have declared that no competing interests exist.

**Abbreviations:** GAPS, Global Assessment of Pediatric Surgery; GDP, Gross domestic product; GICS, Global Initiative for Children's Surgery; IRB, Institutional Review Board; NPAP, Non-physician anesthesia provider; OReCS, Optimal Resources for Children's Surgery; WHO-WFSA, World Health Organization and World Federation of Societies of Anaesthesiologists.

# Introduction

Over 1.7 billion children lack safe and affordable surgical care [1, 2]. Anesthesia is an essential component of providing safe surgical procedures [2, 3]. However, the unique physiological and developmental characteristics of children pose particular challenges for anesthesia care [3–5]. Children require specialized healthcare providers and equipment for ventilation and physiologic monitoring [5–7]. In many nations, limited anesthesia resources lead to a concentration of equipment and trained workforce in urban areas, leaving children living in rural areas without access to safe anesthesia care [7–10]. Identifying where the current gaps of anesthesia care exist, can serve as a first step to strengthen the safe practice of pediatric anesthesia care globally, particularly in nations with limited health systems.

The World Health Organization and World Federation of Societies of Anaesthesiologists (WHO-WFSA) have established standards, guidelines, and specifications to monitor the safe practice of anesthesia care globally [11]. For example, the WFSA Global Anesthesia Workforce Survey highlighted that 66 countries, the majority of them in Africa, still reported a total provider density of less than five in 2024, which is below the recommended goal of at least 5 anesthesiologists per 100,000 population [2, 12]. These recommendations are particularly relevant in settings where local standards do not exist [11, 13]. The standards include the provision of appropriate anesthesia equipment, reliable sources of oxygen and electricity, pediatric-sized equipment, and access to trained anesthesia providers. Similarly, the Global Initiative for Children's Surgery (GICS) has constructed the Optimal Resources for Children's Surgery (OReCS) document, which offers a framework for the expansion of surgical and anesthesia care depending on the hospital level [14].

Our previous work has identified a high burden of pediatric surgical conditions and a lack of surgical resources for children across Somaliland, particularly in rural areas [8, 15–18]. However, pediatric anesthesia capacity across the country is poorly understood. Our study aims to describe anesthesia capacity for children, including workforce, infrastructure, and equipment across Somaliland. Our secondary objectives were to compare these metrics to the WHO-WFSA and OReCS standards and to provide recommendations for scale up of the pediatric anesthesia workforce, infrastructure, and equipment in Somaliland.

# Methods

## Setting

Somaliland is a low-income country located in the Horn of Africa with a gross domestic product (GDP) per capita of $682 in 2018 [19]. The country is divided into six regions: Awdal, Maroodi-Jeex, Sahil, Sanaag, Sool, and Togdheer and has a population of 4.3 million people in 2022 (S1 Fig) [20]. Although Somaliland is not formally recognized as an independent state, it has achieved relative stability as an autonomous government since its separation from Somalia in 1991.

During the conflict with Somalia in 1990, almost all health facilities were damaged or destroyed [21]. Since then, Somaliland has endured a challenging journey to rebuild its healthcare system. The lack of regulations and strategies to train, manage, and finance the health workforce has been a major challenge [22]. Many institutions are unable to provide medical education according to training standards due to the lack of resources and equipment [22]. In addition, the supply chain for medicines and pharmaceuticals is fragmented, leading to low accessibility of essential drugs [22].

Regarding anesthesia care and education, efforts have been made to address the shortage of trained anesthesia providers in the country. In 2013, there was only one formally trained

anesthesia provider in the country [23]. The rest of anesthesia providers had not received formal training. During that same year, the first nurse anesthesia program was started with each class consisting of about twenty students [23].

### Data collection

This cross-sectional study is an extension of our previous work on surgical care for children across Somaliland [16–18]. Data on pediatric anesthesia capacity were collected using a previously validated hospital capacity survey and a pediatric surgical record review tool as described below (S1 and S2 Texts). The surveys were completed between 01 August 2017 and 31 December 2017 and verbal consent was obtained prior to the completion of the surveys. Hospital-based surgical records were reviewed from 01 August 2016 to 31 July 2017.

Hospitals with the capacity to perform surgery, defined as the presence of at least one operating room, were included in the assessment. A total of 15 hospitals out of 16 eligible hospitals were included in this study. The 16th hospital declined participation.

### Hospital capacity survey

At least one anesthesia representative from each hospital was interviewed using a one-page hospital capacity assessment adapted from the Lancet Commission on Global Surgery (*Surgical Assessment Tool-Hospital Walkthrough*) and the Global Initiative for Children's Surgery's *Global Assessment of Pediatric Surgery (GAPS)* [24, 25]. Data included hospital characteristics (location, type, catchment population), workforce, infrastructure, and equipment. The workforce data included number of anesthesiologists and non-physician anesthesia providers (NPAPs). The infrastructure data included the number of operating rooms, oxygen and electricity availability, and time reliance on a generator. The equipment data included the number of functioning ventilators and anesthesia machines. Infrastructure and equipment densities were calculated per 100,000 population and one operating room, respectively.

### Pediatric surgical record review

Surgical records from logbooks were included if the surgical procedure was performed on children between 0 and 15 years old. Data included the type of anesthesia administered and the type of anesthesia providers who performed anesthesia care. Anesthesia providers included anesthesiologists and NPAPs. A total of 1,255 pediatric surgical records were included in the assessment, deidentified, and reported on an aggregate level. Regional workforce density was calculated per 100,000 population, based on regional census data from 2005 and 2014.

### Statistical analysis

Data were reported following the WHO-WFSA standards and OReCS recommendations for the anesthesia workforce, healthcare facilities, and equipment (Table 1) [11, 14]. Other indicators, including medications, intravenous fluids, and monitoring were not included in this study. Descriptive statistics were performed in Microsoft Excel (Version 16.37) and reported in percentages, means, and intervals. The proportion of availability for infrastructure items was reported with median values and ranges for hospital responses. Data from the pediatric surgical record review was used to summarize the type of anesthesia and anesthesia providers across Somaliland. Results were stratified by region.

**Table 1. World Health Organization-World Federation of Societies of Anesthesiologists standards and optimal resources for children's surgical care highly recommended personnel, facilities, and equipment.**

| INDICATOR | WFSA | | OReCS Recommendations |
|---|---|---|---|
| | DEFINITION | RATIONALE/TARGET | RATIONALE/TARGET |
| Anesthesiologist | Graduate of medical school who has completed a nationally recognized specialist anesthesia training program | 5 anesthesiologists per 100,000 catchment population. [a] | Wherever and whenever possible, it is highly recommended that anesthesia delivered to a child should be provided, led or overseen by an anesthesiologist. |
| Nurse anesthetist | Graduate of nursing school who has completed a nationally recognized nurse anesthetist training program | When anesthesiologists are unavailable, it is likely that nurse anesthetists step in to provide anesthesia. | When anesthesia is provided by non-anesthesiologist, these providers should be directed and supervised by anesthesiologists in accordance with their level of training. |
| Non-specialist physician anesthetist | Graduate of medical school who has not completed a specialist training program in anesthesia but has undergone some anesthesia training | When anesthesiologists are unavailable, non-specialist physicians can aid in the provision of anesthesia. | When an anesthesiologist is not present at a specific hospital level, care should be provided by the most experienced anesthesia provider available. |
| Non-anesthesiologist providers | Includes non-specialist physician anesthetists and other providers | Indicator for personnel available in the hospital. | |
| Preoperative area | Dedicated space for preoperative assessment | Suggested but not mandatory. Indicator of anesthesia care quality. | |
| Operating room | Operating rooms are used specially for surgical procedures and are outfitted to deliver anesthesia. | The number of operating rooms available to a population is an indicator for safe anesthesia care. (Global average: 6.2/100,000 persons) [b] | Specifies functional operating rooms and presence of children's ward. |
| Post anesthesia recovery area | Dedicated space for recovering patients. | Recommended but not mandatory. Indicator of anesthesia care quality. | Presence of recovery area. |
| Pediatric anesthesia equipment | Tilting operating table, oxygen, oropharyngeal airways, laryngoscope, adult & pediatric laryngoscope blades, adult & pediatric endotracheal tubes, intubation aids, suction device & catheters, adult & pediatric self-inflating bags, adult & pediatric IVs, spinal anesthesia/regional blocks, defibrillator | This equipment is considered both mandatory and highly recommended for the safe administration of anesthesia. | Size specific anesthesia monitors and equipment should be available at all care levels and facilities in which general anesthesia is provided to allow for children to be adequately monitored. |

[a] The current goal for anesthesia workforces is for there to be 5 anesthesia providers per 100,000 population.

[b] The global average for operating room density is 6.2 operating rooms per 100,000 catchment population. Average across Sub-Saharan Africa is 1.0–1.2 operating rooms per 100,000 population.

## Ethical considerations

This study involves human participants and Institutional Review Board (IRB) approval was granted by Duke University (Duke IRB-Protocol: 2017–0205; E0171). Since Somaliland does not have a national IRB, a formal letter of approval for the study was granted by the Somaliland Ministry of Health. This manuscript adheres to the applicable STROBE guidelines for cross-sectional studies (S2 Checklist).

## Inclusivity in global research

Additional information regarding the ethical, cultural, and scientific considerations specific to inclusivity in global research is included in the (S1 Checklist).

## Results

A total of 15 private were assessed in our study from 16 available hospitals nationwide (the 16th hospital declined participation). Of the 15 hospitals, eight were government hospitals, five

**Table 2. Hospital anesthesia capacity and pediatric surgical procedures (N = 1255) stratified by region and total.** Calculations were made using upper limits of catchment populations, except for Maroodi-Jeex.

| | Total n (%) | Awdal n (%) | Maroodi-Jeex n (%) | Sahil n (%) | Sanaag n (%) | Sool n (%) | Toghdeer n (%) |
|---|---|---|---|---|---|---|---|
| **HOSPITAL ANESTHESIA CAPACITY** [a] | | | | | | | |
| Catchment population | 4,000,000 | 400,000 | 83,000–4,000,000 | 20,00–250,000 | 250,000 | 150,000 | 400,000 |
| Number of hospitals | 15 (100.0) | 3 (20.0) | 6 (40.0) | 2 (13.3) | 1(6.7) | 1(6.7) | 2(13.3) |
| **Workforce** [b] | | | | | | | |
| Anesthesiologists | 3 (100.0) | 0 (0.0) | 2 (66.7) | 0(0.0) | 1(33.3) | 0(0.0) | 0(0.0) |
| NPAPs | 43 (100.0) | 7(16.3) | 22(51.2) | 3(7.0) | 2(4.6) | 1(2.3) | 8(18.6) |
| **Infrastructure** | | | | | | | |
| Operating rooms | 42(100.0) | 8(19.1) | 23(54.8) | 4(9.5) | 1(2.4) | 3(7.1) | 3(7.1) |
| Oxygen (% availability) | (51–75) | (76–99) | (76–100) | (51–99) | (76–99) | (51–75) | (51–99) |
| Electricity (% availability) | (76–100) | (100) | (100) | (76–100) | (76–99) | (100) | (76–100) |
| Time reliance of generator (% availability) | (1–25) | (1–25) | (1–100) | (1–25) | (1–25) | (1–25) | (0–25) |
| **Equipment** | | | | | | | |
| Anesthesia machines | 34 (100.0) | 10 (29.4) | 18(53.0) | 2(5.9) | 1(2.9) | 1(2.9) | 2(5.9) |
| Functioning ventilators | 7 (100.0) | 0 (0.0) | 7(100.0) | 0(0.0) | 0(0.0) | 0 (0.0) | 0 (0.0) |
| **Workforce Density** | | | | | | | |
| Anesthesiologists/100,000 persons | 0.075 | 0.0 | 0.050 | 0.0 | 0.40 | 0.0 | 0.0 |
| Anesthesia providers/100,000 persons [c] | 1.2 | 1.8 | 0.60 | 1.2 | 1.2 | 0.67 | 2.0 |
| Anesthesia providers/operating room | 1.1 | 0.88 | 1.0 | 0.75 | 3.0 | 0.33 | 2.7 |
| **Infrastructure and Equipment Density** | | | | | | | |
| Operating rooms/100,000 persons | 1.1 | 2.0 | 0.58 | 1.6 | 0.40 | 2.0 | 0.75 |
| Anesthesia machines/operating room | 0.81 | 1.3 | 0.78 | 0.50 | 1.0 | 0.33 | 0.67 |
| Functioning ventilators/operating room | 0.17 | 0.0 | 0.30 | 0.0 | 0.0 | 0.0 | 0.0 |
| **PEDIATRIC SURGICAL PROCEDURES** [d] | 1255(100.0) | 240 (19.1) | 954 (75.9) | 20 (1.6) | 2 (0.2) | 0 (0.0) | 40 (3.2) |
| **Anesthesia Type** | | | | | | | |
| General | 1069 (100.0) | 231 (21.6) | 817 (76.4) | 3 (0.3) | 1 (0.1) | 0 (0.0) | 17 (1.6) |
| Regional | 39 (100.0) | 1 (2.6) | 35 (89.7) | 1 (2.6) | 0 (0.0) | 0 (0.0) | 2 (5.1) |
| Local | 147 () | 8 (5.4) | 102 (69.4) | 15 (10.2) | 1 (0.7) | 0 (0.0) | 21 (14.3) |
| **Anesthesia Provider (n)** [e] | 1273 (100.0) | 238 (18.7) | 975 (76.6) | 20 (1.6) | 0 (0.0) | 0 (0.0) | 40 (3.1) |
| Anesthesiologists | 151 (100.0) | 0 (0.0) | 150 (99.3) | 1 (0.7) | 0 (0.0) | 0 (0.0) | 0 (0.0) |
| NPAPs | 1122 (100.0) | 238 (21.2) | 825 (74.9) | 19 (17.3) | 0 (0.0) | 0 (0.0) | 21 (3.6) |

[a] Results are from 15 hospitals with surgical capacity who participated in the hospital capacity survey.

[b] There are no pediatric anesthesiologists in Somaliland.

[c] Anesthesia providers include anesthesiologists and NPAPs.

[d] Results are from 1,255 pediatric surgical procedures identified utilizing the pediatric surgical record review.

[e] Multiple selections available in survey, totals may be over or under hospital specific totals. Providers designated "Unknown" were removed from total anesthesia providers.

were private/for-profit hospitals, and two were charity hospitals. Of the 8 government hospitals, one was a national hospital, six were regional hospitals, and one was a district hospital. In 2017 Somaliland had 3 anesthesiologists and 43 NPAPs, resulting in 0.1 anesthesiologists and 1.2 anesthesia providers per 100,000 population (Table 2). There were no practicing pediatric anesthesiologists in the entire country. Maroodi-Jeex and Sanaag were the only regions with anesthesiologists. Sanaag and Sool had the lowest number of NPAPs, with 2 and 1 anesthetists, respectively. Due to its large catchment population, Maroodi-Jeex (0.6 per 100,000 population) had the lowest anesthesia provider density. There are 1.1 operating rooms per 100,000

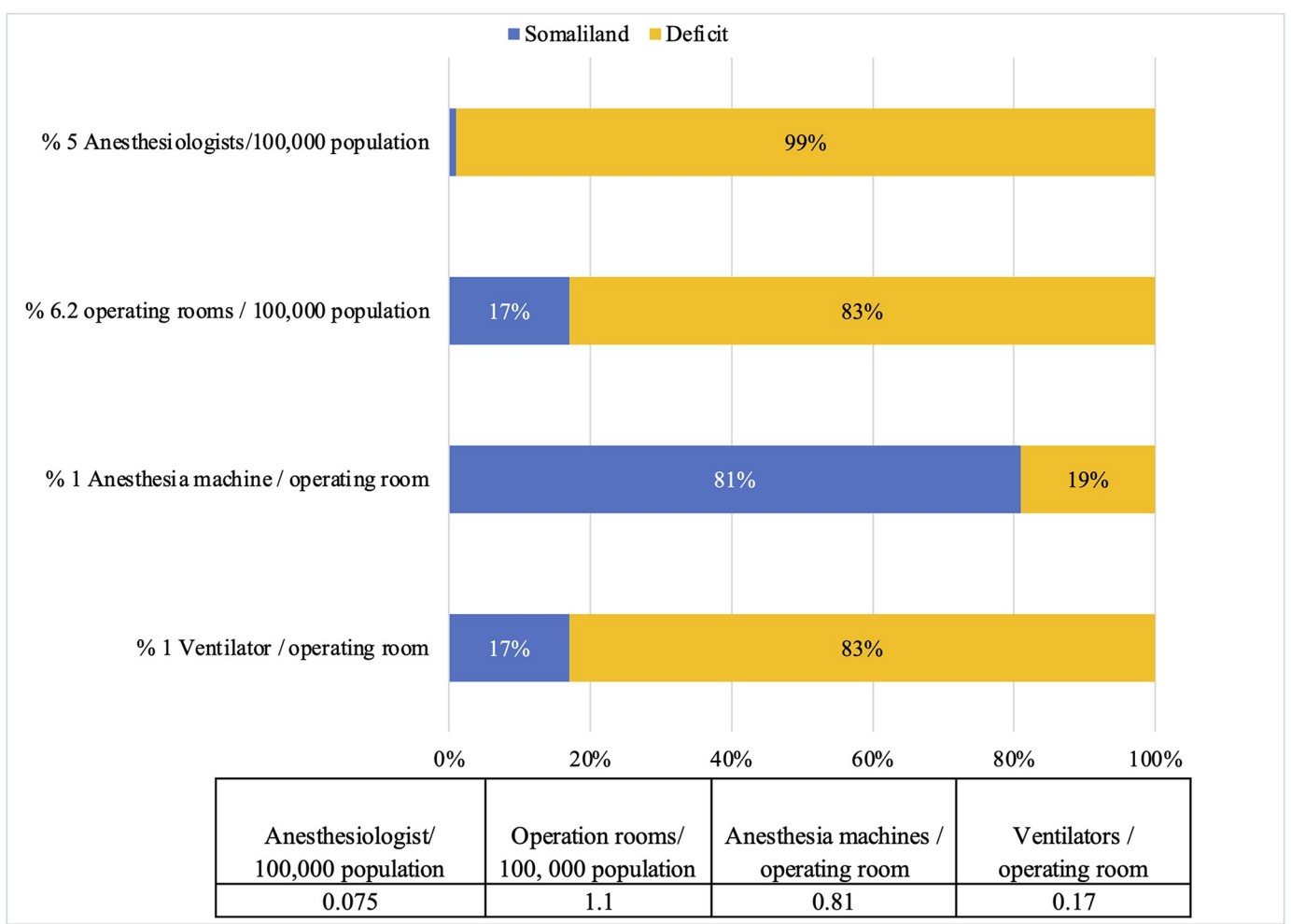

**Fig 1. Percentiles of recommended anesthesia workforce, operating rooms, and anesthesia equipment density in Somaliland.**

population in Somaliland, with the lowest density in Sanaag (0.4 operating rooms per 100,000 population). On average, access to available oxygen ranged from 51–75%, electricity availability ranged from 76–100%, and the time reliance of generators ranged from 1–25%. Maroodi-Jeex had 53% of all available anesthesia machines, while Sanaag and Sool each only had one anesthesia machine. There were seven functioning ventilators, defined as ventilator working correctly and reliably, in the entire country, all of which were located in Maroodi-Jeex.

In 2017 Somaliland had not yet met the target goals for anesthesia capacity defined by the WHO-WFSA and OReCS, with three indicators lagging behind international standards by 83% or more (Fig 1). These three indicators for anesthesia capacity were far behind the established targets, with deficits of 99% for anesthesiologist density, 83% for operating room density, and 83% for ventilator density. In addition, there was a 19% deficit for anesthesia machine density.

From a total of 1,255 pediatric surgical procedures analyzed, 85% (n = 1069) of procedures were conducted under general anesthesia, 3% (n = 39) used regional anesthesia, and 12% used local (n = 147) anesthesia (Table 2). Anesthesiologists were present in 12% (n = 151) of the pediatric operations, and NPAPs were present in 88% (n = 1122) of the pediatric surgical procedures.

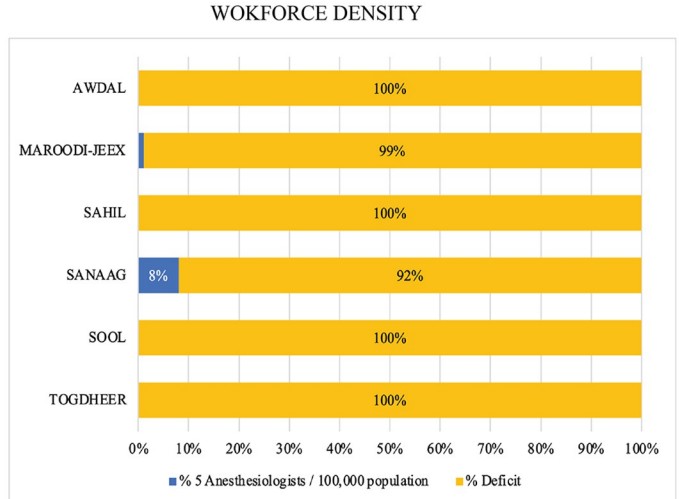

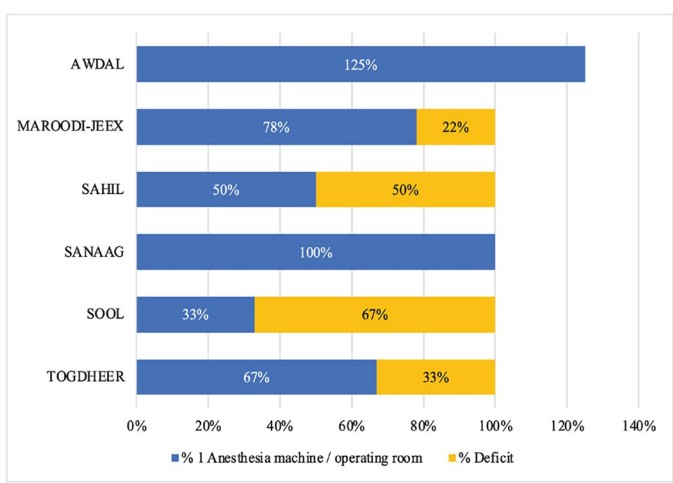

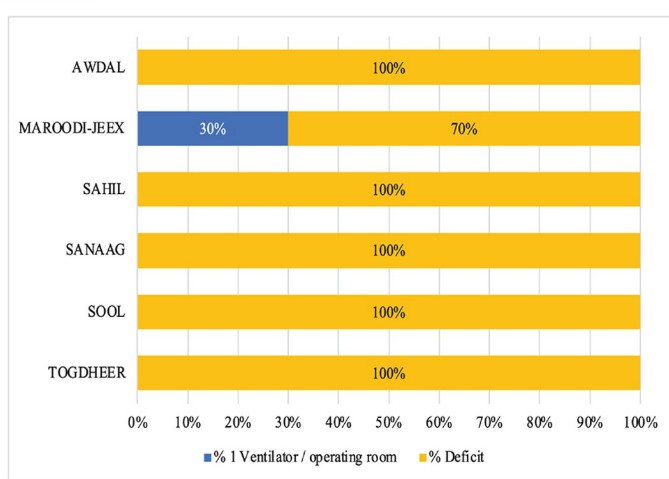

**Fig 2. Percentiles of recommended anesthesia workforce, infrastructure, and equipment density stratified by region in Somaliland.**

Although Sanaag and Maroodi-Jeex had the highest anesthesiologist densities, their deficits from target standards ranged from 92% and 99%, respectively (Fig 2). Awdal and Sool had the lowest deficits for operating room densities (68%), while Sanaag had the highest operating room densities (94%). Maroodi-Jeex had all of the country's ventilators but still had a 70% deficit in comparison to the WHO-WFSA and OReCS standards.

## Discussion

Our results indicate that Somaliland's anesthesia capacity for children achieves only a small fraction of the WHO-WFSA standards and OReCS recommendations, with three indicators lagging far behind international standards, including workforce, operating room, and equipment densities. We found large regional disparities in the anesthesia workforce, infrastructure, and equipment for children's care across Somaliland, with the rural areas of Sool and Sanaag providing only a small fraction of the pediatric anesthesia care in Somaliland.

In Somaliland, 76% of children have unmet surgical needs, with children from rural regions being 4.4 times more likely to have delays in reaching care [8, 15, 16]. These delays may be due

to the lack of pediatric surgical and anesthesia capacity in the rural regions [18]. Families have to embark on long journeys to urban areas like Maroodi Jeex, where more than half of the surgical and anesthesia providers are located. Even after reaching the hospitals in urban areas, timely surgical and anesthesia care is not guaranteed [15]. Many families do not have the financial means to pay for the surgical procedures and have to delay care while they look for the money. [15] For children, such delays can result in lifelong disability or death. The provision of surgical care is multi-dimensional and requires an adequate surgical and anesthesia workforce, infrastructure, equipment, and pediatric supplies [14]. Thus, the severe lack of anesthesia capacity negatively impacts pediatric surgical care, the health of pediatric patients, and the economies of populations with limited financial resources [2, 26].

There is a critical deficit in the anesthesia workforce throughout Somaliland. Even when nurse anesthetists are included in the overall count of all anesthesia providers, the country only reaches 20% of the global target of anesthesia providers. Moreover, the current workforce may overestimate the actual number of anesthesia providers, as many of these providers work at more than one health facility and therefore may be counted more than once in the data. In Somaliland, anesthetists, nurses, or clinical assistants with limited pediatric-specific training provide the majority of anesthesia care. Since over 85% of pediatric surgical procedures in Somaliland were performed under general anesthesia, it can be inferred that the vast majority of general anesthesia in children is not being performed by anesthesiologists and no general anesthesia in children is being performed by pediatric-trained anesthesiologists. Although task-sharing to support a strong anesthesia workforce is essential in many nations with limited healthcare resources, the lack of pediatric-trained providers may put children at increased risk of perioperative complications and negatively impact the quality of surgical care [27].

In addition to workforce limitations, there is a gap in surgical infrastructure across Somaliland. Somaliland has approximately 1.1 operating rooms per 100,000 people compared to the global average of 6.2 operating rooms per 100,000 population. Somaliland's operating room density is consistent with the average of 1–1.2 operating rooms per 100,000 population in sub-Saharan Africa [28]. Operating room density is an important indicator for a population's access to surgical care. Meanwhile, more than 80% of the operating rooms in Somaliland do not have access to a ventilator. When ventilators are not available, hand ventilation is often conducted. However, hand ventilation increases the risk of intra- and post-operative pulmonary complications [5]. In order to meet the burden of surgical disease in children, the GICS through the OReCS document recommends that increased training, infrastructure, and safe modes of anesthesia delivery are developed to provide adequate anesthesia care to children in low-resource settings [14].

Surgical care for children, including anesthesia care, is a cost-effective tool to promote individual children's health and the health of the country [29]. One way to increase pediatric anesthesia care is through the creation of specialized programs focused on pediatric anesthesia for NPAPs or fellowship training for anesthesiologists. In addition, training can be supported by programs such as the WFSA Safer Anaesthesia from Education Paediatric Course (SAFE Paeds), implemented between 2022 and 2003, after our data was collected. So far, 50 anesthesia providers across the country have participated in the course, which has served as a refresher on key knowledge and skills [30]. Although many more providers are needed, this is a start to improving pediatric anesthesia skills in the country.

## Limitations

Our study has several limitations. As a surgical record review, we were limited in our ability to assess the extent of pediatric surgery and anesthesia provided. For example, we could not

determine the type of employment (full-time or part-time), if personnel worked at multiple hospitals, the quality of equipment, available medications for the administration of anesthesia, or supplies for monitoring anesthesia care. For instance, full-time and part-time employment are useful information to determine whether personnel worked at multiple hospitals. If so, the number of anesthesia providers by hospital might inflated. Part-time work would also highlight discontinued access to anesthesia care in some hospitals.

Furthermore, the quality and specific characteristics of anesthesia equipment, such as anesthesia machines and ventilators, are unknown (i.e., stand-alone versus ventilators that are part of an anesthetic machine or whether the ventilators and anesthesia machines had the capacity to serve neonates). This is because reported data included only relative access to equipment and absolute numbers of equipment in hospitals. Finally, although generalizable nationwide, our result cannot be generalizable to other neighboring countries.

## Recommendations

There needs to be 197 more anesthesiologists in Somaliland to meet the current WHO-WFSA standards. To increase the quality of care provided for children, increased pediatric-specific training and resources should be provided for anesthesia providers in rural areas, such as the ongoing SAFE Paeds course [30]. In addition to increasing the number of anesthesia providers in Somaliland, there is also a need for improving anesthesia infrastructure. To come closer to the global average of about 6 operating rooms per 100,000 persons, there is a need to scale up for 198 more operating rooms, 206 more anesthesia machines, and 233 ventilators throughout Somaliland. These efforts should be focused on Sahil, Sool, and Sanaag.

## Conclusion

Safe anesthesia care is essential to surgical care for children. This study has demonstrated large disparities in the number and distributions of anesthesia workforce, infrastructure, and equipment across Somaliland. The scale-up of anesthesia capacity for children should be focused in rural areas with an increase in the number of providers, operating rooms, and equipment in Somaliland. Proportions of the anesthesia workforce and equipment specific to children should reflect the regional and local population needs.

## Supporting information

**S1 Fig. Hospitals included in this study (n = 15).**
(PNG)

**S1 Text. Hospital capacity assessment.**
(DOCX)

**S2 Text. Surgical record tool.**
(DOCX)

**S1 Checklist. Inclusivity in global research.**
(DOCX)

**S2 Checklist. STROBE statement.** Checklist of items that should be included in reports of cross-sectional studies.
(DOC)

## Author Contributions

**Conceptualization:** Mubarak Mohamed, Tessa L. Concepcion, Shukri Dahir, Edna Adan Ismail, Henry E. Rice, Emily R. Smith.

**Data curation:** Mubarak Mohamed, Tessa L. Concepcion, Shukri Dahir, Emily R. Smith.

**Formal analysis:** Andie Grimm, Christina Williams, Cesia Cotache-Condor.

**Investigation:** Mubarak Mohamed, Tessa L. Concepcion, Shukri Dahir, Edna Adan Ismail, Henry E. Rice, Emily R. Smith.

**Writing – original draft:** Mubarak Mohamed, Andie Grimm, Christina Williams, Cesia Cotache-Condor, Tessa L. Concepcion, Shukri Dahir, Edna Adan Ismail, Henry E. Rice, Emily R. Smith.

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
