## [Decision Letter · Decision Letter 0]

29 Apr 2024

PGPH-D-23-02129

Assessment of Anesthesia Capacity for Children in Somaliland

Dear Dr. Smith,

Thank you for submitting your manuscript to PLOS Global Public Health. After careful consideration, we feel that it does not meet PLOS Global Public Health’s publication criteria as it currently stands. Therefore, we invite you to submit a significantly revised version of the manuscript that addresses the points raised during the review process.

My own comments are included below, but mostly importantly I need to know to what extent this data overlaps with an existing publication by some of the same authors, exactly what data collection tool was used and presentation of the full data set (even if this is in appendices).

We look forward to receiving your revised manuscript.

Kind regards,

M. Dylan Bould

Academic Editor

Journal Requirements:

Additional Editor Comments (if provided):

You need to include the actual survey tool that you used in this study as an appendix. You need to explain the process for adapting existing tools and the justification for this.

How did you identify the ''one anesthesia representative" interviewed for each site?

It is important to clarify to what extent the data overlaps with the previous paper https://pubmed.ncbi.nlm.nih.gov/31297580/

The results presented are really quite superficial and certainly do not include all the factors that need to be considered when assessing capacity for providing pediatric anesthesia. Do you have other data from your tool that can be presented as supplementary materials.

Please avoid the unreflective use of the term 'low and middle income countries/LMIC" - see https://www.ncbi.nlm.nih.gov/pmc/articles/PMC9185389/ - at best it is terminology that dichotomizes the world, including a very heterogenous group of countries (and contexts within countries) within a single category.

Reviewers' comments:

Reviewer's Responses to Questions

**Comments to the Author**

1. Does this manuscript meet PLOS Global Public Health’s publication criteria? Is the manuscript technically sound, and do the data support the conclusions? The manuscript must describe methodologically and ethically rigorous research with conclusions that are appropriately drawn based on the data presented.

Reviewer #1: Yes

Reviewer #2: Yes

2. Has the statistical analysis been performed appropriately and rigorously?

Reviewer #1: Yes

Reviewer #2: Yes

3. Have the authors made all data underlying the findings in their manuscript fully available (please refer to the Data Availability Statement at the start of the manuscript PDF file)?

Reviewer #1: Yes

Reviewer #2: Yes

4. Is the manuscript presented in an intelligible fashion and written in standard English?

Reviewer #1: Yes

Reviewer #2: Yes

5. Review Comments to the Author

Reviewer #1: • Thank you for the opportunity to review this paper and apologies for the slow review.

• I have read with interest the manuscript entitled, ‘Assessment of Anesthesia Capacity for Children in Somaliland’. The article is written with the purpose of describing the anesthesia capacity for children, including workforce, infrastructure, and equipment across Somaliland. This evaluation is intended to serve as a foundation for making recommendations to scale up pediatric anesthesia capacity in Somaliland.

• The manuscript is generally well-written and organized, making it easy to follow the flow of ideas. Addressing the suggested revisions outlined below would further enhance the quality and impact of the study.

• Abstract:

The article describes the anesthesia capacity for children in Somaliland; however, the abstract seems to suggest the aim is to describe anesthesia capacity in Somaliland (line 43). It would do well to stick to the aim as stated in the title. This makes it a unique paper and a great addition to the literature on anesthesia capacity for children in LMICs.

• Introduction:

The WFSA Global Anesthesia workforce study referred to in (line 82) has been updated. It might give us a good representation of the state compared to the global status. The Global Anesthesia Workforce Survey: Updates and Trends in the Anesthesia Workforce. Anesthesia and Analgesia. https://doi.org/10.1213/ANE.0000000000006836 . This could enrich the context of the study and its relevance to broader anesthesia workforce trends.

• Methods (Data Collection):

In line 126, you state “previously validated hospital capacity survey” but no reference is provided to this. At least as an attachment (supplementary material) will be useful to make the methods section clearer.

• Results:

Stating the breakdown of the categories of hospitals will provide more information about access to surgical care rather than lumping them up into one basket of “A total of 15 private, charity, and government hospitals” (line 170). If the majority of these hospitals were private, it would reveal even greater challenges regarding access to surgical care for children.

Were the 7 functional ventilators in (line 182) having the capacity to ventilate neonates? It would be good to state that since you interacted with at least one anesthesia provider from each hospital. It is not uncommon to find ventilators that are only able to ventilate adults and maybe big children (teenagers) in low-income countries. The same applies to anesthesia machines (line 181) because having a functional anesthesia machine is different from having one that can take care of neonates. This would add to the magnitude of the existing challenges with anesthesia capacity. Maybe provide the operational definition of a functional ventilator as per this study.

Additionally, were these ventilators part of the anesthesia machines or independent ICU ventilators? Stating this would give us a better description of the current state.

(Line 210) Consider providing an age disaggregation of surgical cases (e.g., neonates, 1 month to 12 months, 1 year to 5 years, and 6 years to 12 years) to contextualize anesthesia capacity relative to different age groups.

Thank you for your attention to these suggestions. Addressing these points will enhance the manuscript's clarity, accuracy, and contribution to the field.

Reviewer #2: Abstract – there is no referencing in the abstract but there should be as it brings up WHO-WFSA standard measurement

Suggest considering the addition of a map of Somaliland with the regions outlined and major towns denoted – even in an appendix for those who are not familiar with the country to see better.

Line 73/74. Consider refashioning the sentence especially with removing the double use of specialized in the same sentence. Suggest rearranging final part ‘ a specialized workforce training’ into a slightly easier to digest phrase.

Suggest re-reading the article with particular attention paid to spacing at the end of a sentence and referencing e.g. lines 77,82, 92, 108, 156 as a few examples where this isn’t done.

Line 107 – lose ‘s’ in millions and consider addition of people after million.

Line 113 – suggest addition of ‘the’ between finance and workforce.

Data Collection.

States that previous data from interviews with anaesthetic providers between 1st Aug 2017 and 31st Dec 2017. This coincides with the paper referenced 19, which relates to collating data from surgical log books AUGUST 2016 – JULY 2017. It states that there were 1255 operations looked at during this time frame and was analysed. This is the same number as stated in the table and results and I would assume that this is the same data. As such, the interviews with anaesthetic providers about provisions and supplies would have taken place after the data collection of the log books and as such there may be some bias of recall of provisions and supplies for the data collected as a different time. But as the data from August 2016-July 2017 is being used this should be included in the data collection prose. But also it does raise the question of why it has taken a length of time to analyse the anaesthetic component of the data from 7 years ago, when the surgical ones were published in 2020.

There is some cross over of the data in the provision of surgical care for children with reference to paper 19 by the same group referenced during the discussion

Line 147 + 364– the demographics states children 0-15y only looked at which is fine. It is a curious number however. Is 15y the age in Somaliland of becoming an adult? Or is it 15 years and 364 days?

Line 150 and table 2. 1255 surgical records of paediatric operations. No mention of any that might have been excluded or proportion of the overall workload.

Stats analysis:

Line 157 – conduct of anaesthesia was excluded but in line 148 it states inclusion of type of anaesthetic administered and line 161 the type of anaesthetic is included.

Results

Table 2.

I am confused with the total number of catchment area being 4,000,000. But it is stated calculations are using the upper limit of the catchment areas apart from Maroodi-Jeex (but doesn’t state what’s used for this region). But also adding up each region’s capacity the total number stated at 4,000,000 doesn’t add up to the total. (and in fact Maroodi-Jeex is stated at 83-4,000,000 and if upper limit used would exceed the total considerably). Suggest checking through the table and the mathematical analysis again.

I am also confused with the data set out regarding anaesthetic procedure cover.

It suggests that 150 operations were provided by an anaesthesiolist in M-J and 1 in Sahil. And yet when looking at where the anaesthesiologists are stated to reside. 2 are in M-J and one in Sanang and 0 in Sahil – so this seems strange. A typo or mislaid column or a visiting anaesthetist?

Also a question – there are more anaesthetic providers present than there are operations conducted (1273 rather than 1255 procedures) . This would seem to suggest that in some operations, more than one provider is present. This would be useful to highlight if so. In lines 212-3 it states there are anaesthetists present in 12% of operations and NPAP in 88%. If it is that there are anaesthetic providers doubled up in some operations then those percentages are inflated and if not more than one anaesthetic provider present then there is a mismatch in the data.

Line 225 It highlights regional discrepancies in delivery of anaesthetic care – but it was only meant to be looking at paediatric anaesthetic care, so it seems a that this cannot be commented on accurately.

Line 182 – only 7 ventilators in the whole country and Line 256 that 80% of operating rooms are without ventilators. I think it needs to be clarified whether we are looking at a mixture of ventilators (stand alone) and/or ventilators as a part of the anaesthetic machine. In table 2 they are highlighted as a separate entity to the anaesthetic machine. There may be some instances with a stand alone ventilator e.g. icu as opposed to an anaesthetic machine ventilator. It would be good to be clear on whether we are looking at functioning ventilators as a part of the anaesthetic machine delivery.

Line 274 – what relevance is full or part time employment?

Line 268 – 50 trained with SAFE courses – these have only been run since 2022-3 thus though good to highlight how things are changing, it wasn’t relevant at that time or needs to be highlighted as something that’s been instituted since the data was collected. Which then should be underlined in line 287 which alludes to the need to deliver SAFE courses regionally – maybe should state that this has already started?

Limitations: I think it is a really unfortunate part of the study that the actual numbers of anaesthetic providers at this time isn’t clear. By own admission there may be cross site working and the number recorded here might be inflated.

6. PLOS authors have the option to publish the peer review history of their article (what does this mean?). If published, this will include your full peer review and any attached files.

**Do you want your identity to be public for this peer review?** For information about this choice, including consent withdrawal, please see our Privacy Policy.

Reviewer #1: **Yes: **Dr. Ronald Bisegerwa

Reviewer #2: No

---

## [Decision Letter · Decision Letter 1]

6 Aug 2024

Assessment of Anesthesia Capacity for Children in Somaliland

PGPH-D-23-02129R1

Dear Dr Smith,

We are pleased to inform you that your manuscript 'Assessment of Anesthesia Capacity for Children in Somaliland' has been provisionally accepted for publication in PLOS Global Public Health.

Best regards,

Barnabas Tobi Alayande

Academic Editor

Thank you for thoroughly addressing the feedback from both reviewers on your manuscript on pediatric anesthesia capacity in Somaliland. All comments have been appropriately addressed from the perspective of an original reviewer and the academic editor. Thank you for all your collaborative work in Somaliland.

Reviewer Comments (if any, and for reference):

Reviewer's Responses to Questions

**Comments to the Author**

1. If the authors have adequately addressed your comments raised in a previous round of review and you feel that this manuscript is now acceptable for publication, you may indicate that here to bypass the “Comments to the Author” section, enter your conflict of interest statement in the “Confidential to Editor” section, and submit your "Accept" recommendation.

Reviewer #1: All comments have been addressed

2. Does this manuscript meet PLOS Global Public Health’s publication criteria? Is the manuscript technically sound, and do the data support the conclusions? The manuscript must describe methodologically and ethically rigorous research with conclusions that are appropriately drawn based on the data presented.

Reviewer #1: Yes

3. Has the statistical analysis been performed appropriately and rigorously?

Reviewer #1: Yes

4. Have the authors made all data underlying the findings in their manuscript fully available (please refer to the Data Availability Statement at the start of the manuscript PDF file)?

Reviewer #1: Yes

5. Is the manuscript presented in an intelligible fashion and written in standard English?

Reviewer #1: Yes

6. Review Comments to the Author

Reviewer #1: All my concerns were addressed appropriately.

7. PLOS authors have the option to publish the peer review history of their article (what does this mean?). If published, this will include your full peer review and any attached files.

**Do you want your identity to be public for this peer review?** For information about this choice, including consent withdrawal, please see our Privacy Policy.

Reviewer #1: **Yes: **Ronald Bisegerwa
